# Evaluation of the Effectiveness of Functional Chewing Training Compared with Standard Treatment in a Population of Children with Cerebral Palsy: A Systematic Review of Randomized Controlled Trials

**DOI:** 10.3390/children9121876

**Published:** 2022-11-30

**Authors:** Alessandra Banzato, Antonella Cerchiari, Sofia Pezzola, Michela Ranucci, Eleonora Scarfò, Anna Berardi, Marco Tofani, Giovanni Galeoto

**Affiliations:** 1Human Anatomy, Histology, Forensic Medicine and Orthopedics, Sapienza University of Rome, 00185 Rome, Italy; 2Department of Intensive Neurorehabilitation and Robotics, Bambino Gesù Children’s Hospital, IRCCS, 00165 Rome, Italy; 3Department of Human Neurosciences, Sapienza University of Rome, 00185 Rome, Italy; 4Professional Development, Continuous Education and Research Service, Bambino Gesù Children’s Hospital, IRCCS, 00165 Rome, Italy; 5Neuromed, IRCCS, 86077 Pozzilli, Italy

**Keywords:** cerebral palsy, chewing treatment, functional chewing, mastication treatment, oral motor exercises, systematic review

## Abstract

Background: Functional Chewing Training (FuCT) was designed as a holistic approach to improve chewing function by providing postural alignment, sensory and motor training, and food and environmental adjustments. The aim of this systematic review was to evaluate the effectiveness of FuCT in improving chewing function and the severity of tongue thrust and drooling in children with cerebral palsy as compared with standard treatment. Methods: We conducted a systematic review of randomized controlled trials. The search was performed between October 2021 and January 2022 using the following databases: PubMed, Scopus, Web of Science, and CINAHL. The review was performed according to Preferred Reporting Items for Systematic Reviews and Meta-Analyses (PRISMA) guidelines. Results: The initial search yielded 56 articles. After reading the studies in full, 3 articles were chosen based on the inclusion criteria. Included participants were people with PCI; the studies reported a sample size ranging from 40–80 individuals, one study was on a pediatric population, while the others on adults. The selected studies were then evaluated using Jadad and PEDro scales. Conclusion: Our study confirmed the value of FuCT in improving chewing function and the severity of tongue thrust and drooling. Our results may be useful in optimizing appropriate therapeutic management.

## 1. Introduction

Cerebral palsy (CP) is a heterogeneous group of nonprogressive motor disorders caused by chronic brain injuries that originate in the perinatal period or during the first few years of life [1]. Children with CP have various feeding and swallowing difficulties, including chewing dysfunction, which may affect their nutritional status, growth, and quality of life.

The prevalence of nutritional problems in children with CP ranges from 30% to 90%, with malnutrition seen in 90% of this patient population [2]. The potential reasons for nutritional problems in children with CP include oral motor dysfunction, postural problems, persistence of primitive reflexes, chewing disorders, drooling, and gastrointestinal problems [3,4].

Feeding behavior is considered a sensitive indicator of central nervous system (CNS) integrity in neonates. The need for nasogastric feeding in babies born at term may thus indicate a subtle developmental abnormality of the CNS. Primitive reflexes consist of involuntary motor responses originating in the brainstem that are present after birth and during early childhood development and that facilitate survival [5].

When CNS development is completed in normally developing children, primitive reflexes are inhibited and more specific and voluntary movements can be performed [6,7]. The persistence of five or more abnormal reflexes is correlated with the development of CP or mental delays and affects the development of related functions [8,9]. Tongue thrust is an oral reflex associated with sucking behavior during infancy. If sucking behavior cannot be suppressed after six months of life, it is referred to as tongue thrust; the continuation of this pattern may cause problems in swallowing, speech, and orofacial development, and may also cause drooling [10,11]. Due to its negative consequences, affected children and their families may experience social and emotional problems [12].

The presence of oral motor dysfunction is highly represented in the population of children with CP. According to some studies, more than 90% of preschool children with CP have clinically significant oral motor dysfunction [9]. In general, the more severe the functional motor impairment, the more severe the oral motor dysfunction [3]. Difficulties in oral motricity may influence the ability to chew, swallow, and control salivation.

Drooling is one of the most frequent problems seen in children with CP. Drooling is defined as the loss of saliva from the mouth. The prevalence of drooling in children with CP ranges from 37.4 to 58%. It is related to oral phase dysfunction in addition to insufficient lip closure generally and to tongue movements that are impaired due to diminished oral and perioral sensory perception, upside-down posture, diminished swallowing frequency, and dysphagia. Some studies have also found that difficulties in swallowing saliva are related to postural abilities, such as head control [13,14]. Moreover, the inability to swallow saliva can lead to aspiration pneumonia [15].

Chewing function, which is part of the feeding process, is defined as a series of rhythmic oral motor activities that include biting, lateral and rotational tongue movements, even elevation and retraction of the tongue, and swallowing, which are necessary to comminute and soften solid food [12]. Children with CP often have difficulty with bolus formation and effective chewing and have limited ability to manage age-appropriate food textures [9]. The specific nature and severity of the chewing dysfunction may differ in relation to sensorimotor impairment and gross and fine motor limitations [14]. The most affected aspects of chewing are food transportation from the front of the mouth to the molar area and food processing through masticatory cycles due to insufficient lateral and rotational tongue movements [16]. Thus, children with chewing dysfunction are unable to take any solid food, while the diet of normally developing children with normal feeding skills includes liquid, semisolid, and/or solid foods together [8,16]. This inability may limit sufficient food intake and the nutritional status of children. Nutritional status in turn affects the growth, general health, and quality of life of children and their families [17].

Therefore, it is important to improve chewing function and solid food intake in children with CP. Oral motor programs [13] are used to train orofacial structures to improve sensory integration, motor coordination, and muscular strength. To improve swallowing and chewing, oral motor programs aim to increase the tactile and proprioceptive aspect of eating. Oral motor interventions aim to improve mouth function and control by gradually thickening the texture of foods and teaching families proper positioning [15]. Recently, a new approach was developed: the Functional Chewing Training (FuCT) [18]. FuCT was designed as a holistic approach to improve chewing function by providing postural alignment, sensory and motor training, and food and environmental adjustments. Therefore, it may also be used to reduce tongue thrust in children with CP. The philosophy of this training method is that chewing is a learned behavior, and therefore repeated positive and successful experiences are key in learning how to chew through FuCT, which incorporates therapy sessions and daily rules [19]. Although some published studies have aimed to investigate the effectiveness of this method, there are no known clinical practice guidelines that currently support FuCT interventions in this population. This approach has been developed specifically for children with PC in 2017 [18], to date, it has also been studied in children with repaired Esophageal Atresia and Tracheoesophageal Fistula (EA/TEF) [20]; moreover, in the literature, it is possible to find evidence form Turkey and China [21].

The primary purpose of the present paper was to evaluate the efficacy of FuCT on chewing function and related components, including the severity of tongue thrust and drooling, in a population of children with CP through the analysis of randomized controlled trials (RCTs).

## 2. Methods

This systematic review was conducted according to the principles of Preferred Reporting Items for Systematic Reviews and Meta-Analyses (PRISMA) guidelines [22,23] by a research group of Sapienza University of Rome and the Rehabilitation & Outcome Measures Assessment (ROMA) Association, who were involved in different studies on rehabilitation [24,25,26,27,28,29,30,31,32], The study did not require human participation, and therefore ethics approval was not required.

### 2.1. Inclusion and Exclusion Criteria

Studies were included if they were RCTs that reported the efficacy of FuCT in improving chewing function and the severity of tongue thrust and drooling in children with CP. No study was excluded based on language or publication date due to the lack of published systematic reviews on this topic. The reference population of the included studies was children diagnosed with CP and aged between 0 and 18 years.

### 2.2. Data Source and Search Strategy

A systematic literature search of articles published between October 2021 and January 2022 was performed using the following electronic databases: PubMed, CINAHL, Scopus, and Web of Science. Search terms were designed to include the population of interest (‘cerebral palsy (MeSH)) AND intervention type (‘Functional Chewing Training’ OR ‘functional chewing treatment’ OR ‘functional mastication treatment’ OR ‘FuCT’ OR ‘functional mastication training’) AND study design (‘randomized controlled trial’) (Table 1).

Five of the authors (AB, AC, ES, MR, and SP) independently examined the titles and abstracts of the articles identified by these searches. Full-text articles were retrieved if they fulfilled the inclusion criteria, or if further clarification regarding the fulfilment of inclusion criteria was required. If agreement on inclusion could not be reached following review by five of the authors, the sixth author (GG) was consulted.

### 2.3. Data Extraction

A table was developed to summarize the data extracted from the selected studies, including the following information: (1) references (authors and year of publication); (2) participant characteristics (number, age, gender, and motor function level); (3) intervention; (4) control (type, duration, and frequency); (5) outcome measure; and (6) results.

### 2.4. Risk of Bias

To assess the quality of studies, Jadad [33,34] and PEDro [35] scores were calculated for each study. The Jadad score considers key aspects of a high-quality trial: randomization, blinding, and subjects lost to follow-up. The PEDro scale is based on the following items:Eligibility criteria were specified;Subjects were randomly allocated to groups (in crossover studies, subjects were randomly allocated to an order in which treatments were received);Allocation was concealed;The groups were similar at baseline regarding the most important prognostic indicators;All subjects were blinded;All therapists who administered the therapy were blinded;All assessors who measured at least one key outcome were blinded;Measures of at least one key outcome were obtained from more than 85% of the subjects initially allocated to groups;All subjects for whom outcome measures were available received the treatment or control condition as allocated or, when this was not the case, data for at least one key outcome were analyzed by ‘intention to treat’;The results of between-group statistical comparisons were reported for at least one key outcome.

## 3. Results

### 3.1. Study Selection

A total of 56 articles were identified using the selected search terms. After removing 8 duplicates, 48 reports were screened based on title and abstract. Of these, five articles were initially selected. After reviewing the full text, only three [18,21,36] met the inclusion criteria and were chosen for the review. The flow of studies and reasons for exclusion at each stage are summarized in Figure 1.

### 3.2. Participants

The selected studies included information from 168 patients with CP. The sample size of the included studies ranged from 40 [36] to 80 children [18].

### 3.3. Study Characteristics

Detailed descriptions of characteristics and outcomes of the included trials are presented in Table 2. A total of three studies with 168 total participants were summarized. All trials examined the effects of FuCT on oral abilities in children with CP. The motor level of the participants included was first evaluated in two of the trials [21,36] using the Gross Motor Function Classification System (GMFCS) [37,38,39]. None of the studies reported adverse effects.

### 3.4. Intervention

FuCT was the treatment of interest in all three studies. All studies reported the treatment modality, but only one [18] described all the steps of FuCT in detail:➢Step I (positioning the child);➢Step II (positioning the food);➢Step III (sensory stimulation);➢Step IV (chewing exercises);➢Step V (adjustment of food consistency).

The treatment duration was 12 weeks. The control groups were treated with traditional oral motor exercises that included passive and active lip and tongue exercises. Active exercises included an active range of motion and strength training of the lips and tongue.

### 3.5. Outcome Measures

The rating scales used in the three studies to evaluate outcomes included the following:The Behavioral Pediatrics Feeding Assessment Scale (BPFAS) is a 35-item standardized, reliable, and valid parent-completed screening tool. Each item is rated on a 5-point Likert scale based on the frequency with which particular behaviors occur [40].The Karaduman Chewing Performance Scale (KCPS) is a valid, reliable, quick, and clinically easy-to-use instrument to determine the level of chewing function in children [41].The Tongue Thrust Rating Scale (TTRS) is the first and only scale that is valid, reliable, quick, and clinically easy-to-use to define tongue thrust severity in children [42].The Drooling Severity and Frequency Scale (DSFS) was used to evaluate drooling severity and frequency [43]. Parents were asked to rate the severity and frequency of drooling.

### 3.6. Risk of Bias

The methodological quality of the selected studies was assessed using the Jadad and PEDro scales for each study. Regarding Jadad scores, two studies obtained a final score of 4 [18,36], and one obtained a final score of 2 (Table 3). The PEDro scale considers the description of the key aspects of a high-quality trial in the rehabilitation field; one study obtained a final score of 8 [18], another obtained a final score of 6 [36], and the last obtained a final score of 7 (Table 4).

### 3.7. Effects of FuCT on Chewing, Tongue Thrust and Drooling

All three trials demonstrated an improvement in chewing function after FuCT. Two trials [31,32,36] showed that the FuCT was an efficient method to improve tongue thrust severity and frequency and severity of drooling. In the study of Inal et al. [36], an improvement on the KCPS of 12 points in the intervention group was reported, while the control group reported between t0 and t1 of only 3 points of difference; also, for the other assessment tools, it is possible to notice a high magnitude of improvements. The same result was confirmed by the study of Li Fan et al., which reported an improvement on the KCPS of 12 points in the FuCT group and of 3 points in the control group. In the same assessment tool, Serel Arslan et al. [18], who conducted the study on a pediatric population, found an improvement of more than 100 points in the FuCT group compared with fewer than 10 points. Results are summarized in Table 2.

## 4. Discussion

Currently, there is not much evidence about the effectiveness of FuCT. Searches in four different databases returned only four RCTs, and only three were included in this systematic review. All RCTs on FuCT effectiveness were conducted in recent years (2017–2020). The three different RCTs examined in this review [18,21,36] evaluated the efficacy of FuCT in improving chewing function. In all selected studies, patients randomly assigned to the FuCT intervention group showed improvements in chewing function as compared with the control group. Two studies also investigated the effects of FuCT on chewing, drooling, and tongue thrust, however, no studies have examined the development of oral abilities and improvement in oral dysfunction.

Arslan and colleagues [18] considered only chewing function and observed statistically significant improvements (*p* < 0.001) in chewing performance according to KCPS score in the FuCT group (50 patients) as compared with the control group (30 patients) treated with traditional oral motor exercises.

Inal and colleagues [36] initially included 40 participants (FuCT group = 20; control group = 20), but the study was completed with 16 participants in each group, since 8 were lost to follow-up due to epileptic seizures, Botox application, or surgical intervention. After treatment, improvement in chewing function was statistically significant (*p* = 0.001) in the FuCT group. Statistically significant improvements in tongue thrust severity (*p* = 0.046) and drooling severity (*p* = 0.002) were also found in the FuCT group, but no improvements were found in terms of drooling frequency (*p* = 0.082).

Fan and colleagues [21] observed statistically significant improvements in chewing function, tongue thrust severity, and drooling severity (*p* < 0.050) in the FuCT group (24 patients), but no statistically significant improvements were found in terms of drooling frequency (*p* > 0.05).

All three studies evaluated chewing function using the KCPS, which allowed the results of all three studies to be compared.

The risk of bias, as evaluated by the Jadad and PEDro scales, indicated that most studies had medium-high methodological quality, although one did not report the blinding method [21] and another did not specify losses and exclusions according to intention to treat [36]. Additional research is needed to expand the evidence regarding FuCT in patients with CP.

### 4.1. Limitations

The main limitation of this systematic review is related to the scarce evidence of FuCT use in children with CP. Other limitations are due to characteristics of the included studies, including the small sample size, the superficial description of control treatment (oral motor exercises), and possible methodological biases. It is necessary to conduct studies with a large number of participants so that conclusive results and sufficient scientific evidence are obtained. There is insufficient scientific evidence to compare FuCT effectiveness in children with CP; it was not possible to conduct the meta-analysis because there were no comparable outcomes or comparable follow-ups.

### 4.2. Clinical Implications

This review demonstrates that FuCT may exert beneficial effects on chewing function, severity of tongue thrust, and severity and frequency of drooling in children with CP. This finding may facilitate clinician decision making regarding alternatives to the conventional management of orofacial dysfunction in children with CP.

## 5. Conclusions

Although the amount of evidence in the literature is not yet sufficient, most outcome measures observed in the included studies showed FuCT benefits in terms of chewing dysfunction and tongue thrust and drooling severity. The use of FuCT could support the development of oral abilities and improvement in oral dysfunction. FuCT may be recommended as a valid treatment in the field of speech therapy.

Additional investigations concerning the clinical applicability of this therapy based on well-designed RCTs with a larger sample size are needed, which could help better define FuCT effectiveness and long-term effects. A larger sample would better represent the target population, because it would provide more variables to reason about. Conducting a study that included multiple participants with varying disease severity, ages, etc., would provide researchers with the ability to conduct subgroup analyses in order to give clinicians more accurate information about the outcomes that can be expected for a patient who falls within certain criteria. Even though only one study reported dropout due to epilepsy-associated conditions, with reference to participants lost to follow-up [36], it is recommended to evaluate a priori the condition of subjects with epilepsy associated with CP.

The use of the GMFCS to evaluate motor function level is recommended in future longitudinal trials, but we suggest also to include the Eating and Drinking Ability Classification System [44] to guarantee homogeneity of the sample, both assessment tools demonstrated good psychometric properties, both for validity and reliability [45,46].

## Figures and Tables

**Figure 1 children-09-01876-f001:**
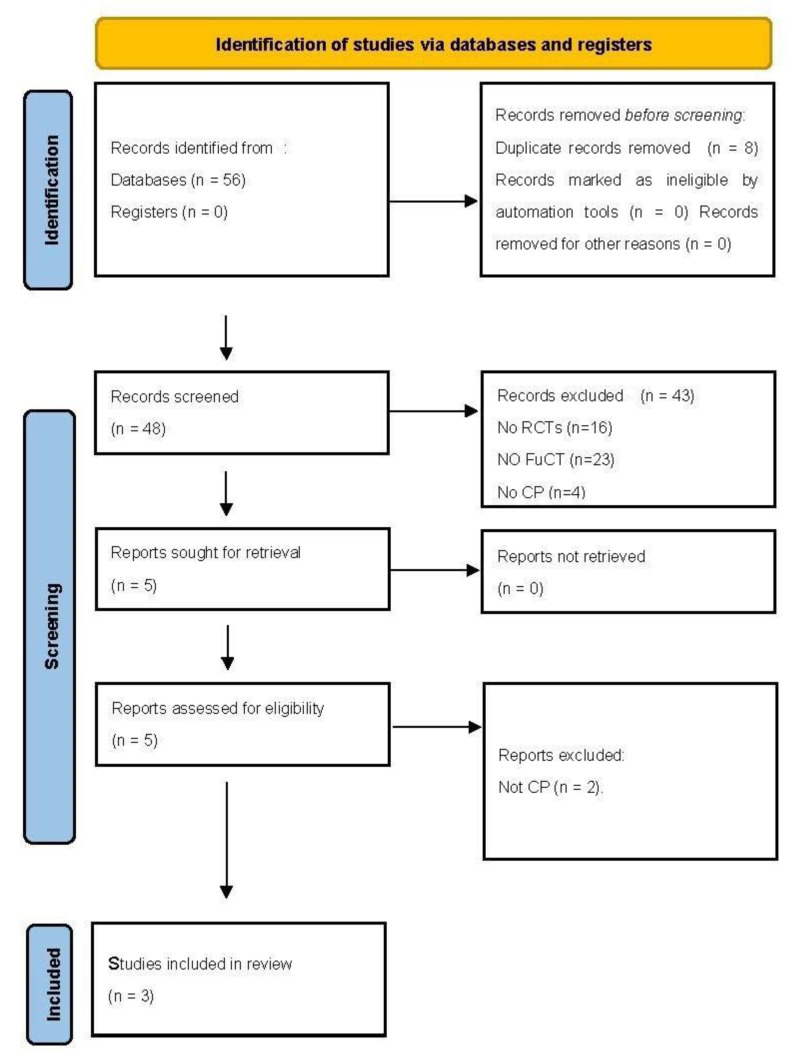
Articles selected from the databases.

**Table 1 children-09-01876-t001:** Search of four databases.

Database Search
PubMed
(Cerebral palsy (MeSH)) AND (‘Functional Chewing Training’ OR ‘functional chewing treatment’ OR ‘functional mastication treatment’ OR ‘FuCT’ OR ‘functional mastication training’)
Scopus
(Cerebral palsy) AND (Functional Chewing Training) OR (functional chewing treatment) OR (functional mastication treatment) OR (FuCT) OR (functional mastication training)
CINAHL
(Cerebral palsy) AND (Functional Chewing Training) OR (functional chewing treatment) OR (functional mastication treatment) OR (FuCT) OR (functional mastication training)
Web of Science
(Cerebral palsy) AND (Functional Chewing Training) OR (functional chewing treatment) OR (functional mastication treatment) OR (FuCT) OR (functional mastication training)

**Table 2 children-09-01876-t002:** Study characteristics and main results.

Author/Year	Participants	Intervention	Control	Outcome Measure	Results
Arslan et al., (2017) [18]	FuCT group	The protocol aimed to ensure functional chewing improvement by stimulating and teaching the function. The FuCT is a holistic approach that includes therapy sessions (steps 1, 3, and 4) and daily rules (steps 1, 2, and 5). It takes 20 min to complete. FuCT was performed with five sets/day and for 5 days a week over a period of 12 weeks as a home program.	The control group received traditional oral motor exercises including passive and active lip and tongue exercises. Passive exercises included range of motion exercises. Exercises were performed with five sets/day and for 5 days a week over a period of 12 weeks as a home program.	Behavioral Pediatrics Feeding Assessment Scale (BPFAS) Karaduman Chewing Performance Scale (KCPS)	After 12 weeks, the FuCT group showed improvement in chewing performance according to the KCPS (*p* < 0.001) and in feeding behaviors according to the BPFAS (*p* < 0.001). A significant improvement was found in the FuCT group as compared with the control group in KCPS score and in all BPFAS subscale scores, except the restriction score after 12 weeks of the intervention (*p* < 0.001).
	*N* = 50				
	*Age* = 3.5 (±1.9) years				
	*Gender* = 19 F/31 M				
	*Motor function level was not specified.*				
	Control group				
	*N* = 30				
	*Age* = 3.4 ± 1.9 years				
	*Gender* = 14 F/16 M				
	*Motor function level was not specified.*				
Inal et al., (2017) [31]	FuCT group	Families were asked to perform FuCT exercises regularly for 12 weeks with five sets (1 set = 20 min) each day.	Group II received a traditional oral motor exercise program. Families were asked to perform the exercises regularly for 12 weeks with five sets (1 set = 20 min) each day.	Gross Motor Function Classification System (GMFCS) Karaduman Chewing Performance Scale (KCPS) Tongue Thrust Rating Scale (TTRS) Drooling Severity and Frequency Scale (DSFS)	After 12 weeks of treatment, the FuCT group showed improvement in chewing performance according to KCPS score (*p* = 0.001), in tongue thrust according to TTRS score (*p* = 0.046), and in drooling severity according to DSFS score (*p* = 0.002), but no improvement was found in terms of drooling frequency (*p* = 0.082).
	*N* = 20				
	*Age* = 43.8 months F/56.2 months M				
	*Gender* = 7 F/9 M				
	*GMFCS* = L1 (0)/L2(1)/L3(4)/L4(0)/L5(11)				
	Control group				
	*N* = 20				
	*Age* = 37.5 months F/62.5 months M				
	*Gender* = 6 F/10 M				
	*GMFCS* = L1(0)/L2(1)/L3(6)/L4(0)/L5(9)				
Fan et al., (2020) [32]	FuCT group	The protocol aimed to improve chewing function, tongue function, and severity and frequency of drooling. Both groups received FuCT or oral motor training for 12 weeks, 5 times a day, and for 10 min each time.	The control group received traditional oral motor exercises. Families were asked to perform the exercises regularly for 12 weeks with five sets (1 set = 20 min) each day.	Gross Motor Function Classification System (GMFCS) Karaduman Chewing Performance Scale (KCPS) Tongue Thrust Rating Scale (TTRS) Drooling Severity and Frequency Scale (DSFS)	After a 12-week training, the FuCT group showed significant improvements in masticatory function, tongue thrust severity, and drooling severity (*p* < 0.05), but no improvement in drooling frequency (*p* >0.05), while the oral motor training group had no improvement in masticatory function, tongue thrust severity, or drooling severity or frequency (*p* > 0.05). After the 12-week training, the FuCT group had more significant improvements in tongue thrust severity and drooling severity and frequency than the oral motor training group (*p* < 0.05).
	*N* = 24				
	*Age* = 5.5 years				
	*Gender* = 11 F/13 M				
	*GMFCS* = L1(1)/L2(3)/L3(6)/L4(2)/L5 (12)				
	Control group				
	*N* = 24				
	*Age* = 5.1 years				
	*Gender* = 8 F/16 M				
	*GMFCS* = L1(1)/L2(4)/L3(5)/L4(2)/L5 (12)				

**Table 3 children-09-01876-t003:** Risk of bias according to the PEDro scale.

Jadad Scale Item			
Author	Randomization	Blinding	Account of Patients
Arslan et al., (2017) 4/5 [18]	2 (Randomized and split between the FuCT group and the control group using randomized sampling, which was computer-generated with a basic random number generator; the allocation ratio was 5:3)	1 (This study was designed as a double-blind RCT of FuCT in patients with CP as compared with traditional oral motor exercises)	1 (Deducted from the tables)
Inal et al., (2017) 4/5 [31]	2 (Of the 40 participants, 20 were randomized to the FuCT group and 20 to the traditional oral motor exercise group with block randomization methods. The Random Allocation Software 2.0 program was used to randomize two groups with an equal number to the block randomization system)	1 (Evaluations were performed in a standardized manner at baseline and after 12 weeks of treatment by an experienced physical therapist blinded to the group allocation of the children)	1 (Flow chart)
Fan et al., (2020) 2/5 [32]	1 (Casual randomization)	0 (Blinding was not mentioned)	1 (Deducted from the tables)

**Table 4 children-09-01876-t004:** Risk of bias according to the PEDro scale.

	PEDro Scale Item											
Author	1	2	3	4	5	6	7	8	9	10	11	Final Result
Arslan et al. (2017) [18]	YES	YES	YES	YES	YES	NO	NO	YES	YES	YES	YES	8/10
Inal et al. (2017) [31]	YES	YES	YES	YES	YES	NO	NO	NO	NO	YES	YES	6/10
Fan et al. (2020) [32]	YES	YES	YES	YES	NO	NO	NO	YES	YES	YES	YES	7/10

## Data Availability

The protocol has been submitted to Prospero database (378978) and it is awaiting approval.

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
