# Peer review of "Evaluation of the Effectiveness of Functional Chewing Training Compared with Standard Treatment in a Population of Children with Cerebral Palsy: A Systematic Review of Randomized Controlled Trials"

_children, 2022, doi:10.3390/children9121876_

Round 1

Reviewer 1 Report

First of all, I want to note that it has been a pleasure review your manuscript. I think this is an interesting work on the Functional Chewing Training in children with Cerebral palsy.

After reading in depth the manuscript, I would like to make some comments:

-         I have been struck by how few studies there are in a review for conclusions to be drawn.

-        Deben poner el mismo tamaño de letra en el título. Please put the same font size in the title.

-         Page 1, line 35, 40... end wrongly. Please revise the whole document.

-        Lines 46, 54…...Correct the reference should go before the full stop. Please revise the whole document.

-        Has the review been registered in any database as PROSPERO? If so, the code received should be entered.

-        The items of the PEDro scale are in a different font size.

-        Page 4, line 161: correct: [Please insert Fig 1 and Table 2]

-        Remove the following sentence from line 160: “Study characteristics are shown in Table 2”. It is already placed in its corresponding part of the lines 162-163.

-        In the results section, please, write  “participants section” first and then the “characteristics of the studies section”.

-        The steps of the intervention are in different sizes. Line 176. Also the scales (line 186)

-        In Risk of bias section: “The PEDro scale considers the description of the key aspects of a high-quality trial in the rehabilitation field. One study 201 obtained a final score of 817 (Arslan et al. 2017). Please delete what is in brackets. The reference has already been put in the proper format before. Correct this paragraph, there are lines that end badly, and spaces that should not be there.

-        A full stop separating the sentences is missing in line 209.

-        The results section should be improved and made more orderly.

-        The discussion section should be improved. It limits itself to explaining results without comparing or discussing them.

-        There are too few studies to conclude the discussion so firmly. Page 7, line 258

-        The conclusions section should be improved in order to meet the objective of the work.

-        Review all references. All authors should be listed, not simply et al. For example:

Gastrostomy tube feeding in children with cerebral palsy: a prospective, longitudinal study

Peter B Sullivan 1Edmund JuszczakAllison M E BachletBridget LambertAngharad Vernon-RobertsHugh W GrantMuftah EltumiLiz McLeanNicola AlderAdrian G Thomas

-        Improve the table 4. Put the final result in a separate column, please.

-        Not all parts of figure 1 are visible. The reasons why items were excluded should be listed.

-        Table 2 should list the variables being measured and then the scales or measurement instruments.

Author Response

Dear Editor,

We appreciate the opportunity to resubmit our article entitled “Evaluation of the effectiveness of Functional Chewing Training compared to standard treatment in a population of children with cerebral palsy: A systematic review of randomized controlled trials.” We would like to thank the referees for the careful and constructive reviews. We have made corresponding changes directly to the manuscript where appropriate with changes tracked. The revised version of our manuscript accompanies this letter. All comments by the reviewer have been addressed. Based on his/her comments, we have made changes to the manuscript, which are detailed below.

Reviewer Comment

Response

Line #

Reviewer #1

I have been struck by how few studies there are in a review for conclusions to be drawn.

That’s true, however, authors believe it is important to highlight how scares is the literature in order to induce research to proceed in this field to find the best evidence.

Please put the same font size in the title.

The font has been corrected by the Editor

Throughout the text

Page 1, line 35, 40... end wrongly. Please revise the whole document.

I apologize but we could not find the sentence cited by the reviewer. However, we revised the whole document and corrected some typos

Throughout the text

Lines 46, 54…...Correct the reference should go before the full stop. Please revise the whole document.

The manuscript has been revised

Throughout the text

Has the review been registered in any database as PROSPERO? If so, the code received should be entered.

Information about the protocol have been added

282-284

The items of the PEDro scale are in a different font size.

The font has been corrected by the Editor

Throughout the text

Page 4, line 161: correct: [Please insert Fig 1 and Table 2]

The manuscript has been corrected by the Editor, and it is not editable I guess

Remove the following sentence from line 160: “Study characteristics are shown in Table 2”. It is already placed in its corresponding part of the lines 162-163.

The sentence has been removed

Line 162

In the results section, please, write  “participants section” first and then the “characteristics of the studies section”.

The section has been moved

Results

The steps of the intervention are in different sizes. Line 176. Also the scales (line 186)

The font has been corrected by the Editor

 In Risk of bias section: “The PEDro scale considers the description of the key aspects of a high-quality trial in the rehabilitation field. One study 201 obtained a final score of 817 (Arslan et al. 2017). Please delete what is in brackets. The reference has already been put in the proper format before. Correct this paragraph, there are lines that end badly, and spaces that should not be there.

This section has been revised

Risk of Bias section

A full stop separating the sentences is missing in line 209.

Full stop has been added

Line 209

The results section should be improved and made more orderly.

Results section has been reported according to PRISMA checklist

Results section

The discussion section should be improved. It limits itself to explaining results without comparing or discussing them.

Discussion section has been improved

Discussion section

 There are too few studies to conclude the discussion so firmly. Page 7, line 258

The sentence has been mitigated

The conclusions section should be improved in order to meet the objective of the work.

Conclusion section has been improved

Conclusion section

Review all references. All authors should be listed, not simply et al. For example:

References has been reviewed

References

Improve the table 4. Put the final result in a separate column, please.

Table 4 has been corrected

Table 4

-        Not all parts of figure 1 are visible. The reasons why items were excluded should be listed.

Figure 1 has been corrected accordingly to reviewer’s comment

figure 1

Table 2 should list the variables being measured and then the scales or measurement instruments.

According to other reviewer in order to make the table more readable we decided not to include the list of variable included, they can be found in the text

Reviewer #2

1. The authors summarized the statistically significant chewing performance improvements in the 3 studies, but did not discuss the magnitude of improvements (is it a very small improvement, or a big enough improvement to make the children's life much better?). The authors should also state the KCPS scores (and other scores) in trial and control groups, and what the change means for the patient's life quality. 2. Does the magnitude change differ across the 3 studies?

magnitude of improvements has been reported

213-221

1. The authors stated "It is necessary to conduct studies with a large number of participants so that conclusive results and sufficient scientific evidence are obtained." Why do we need larger-size studies when 3 independent studies already give consistent and statistically significant conclusions? I would argue that even with some limitations of the previous work, 3 medium-to-high-quality studies (as evaluated by Jadad and PEDro) showing the same result means we don't need to spend more funding resources to repeat the same kind of study. Regarding the limitations of the 3 studies, small sample size is not an issue if the study can draw statistically significant conclusions (i.e. it already has sufficient statistical power).

A larger sample would better represent the target population because it would provide more variables to reason about. Conducting a study that included multiple participants with varying disease severity, ages, etc. would provide researchers with the ability to do subgroup analyzes in order to give clinicians more accurate information about the out-comes that can be expected for a patient who falls within certain criteria.

268-273

2. The statement "There is insufficient scientific evidence to compare FuCT effectiveness in children with CP." is not consistent with the result of the meta-analysis.

It was not possible to do the meta-analysis because there were no comparable outcomes or comparable follow-ups

268-269

3. The authors should include demographic information of the 3 studies and their participants.

Demographic characteristics of participants are reported in table 2

Reviewer #3

My major concern is that the results of this review have not been thoroughly discussed. And recommendations for future study are too brief and lack strong justification.

Tish section has been revised

Discussion section

Abstract:

1.         Instead of mentioning how many studies were screened and removed in each step in the results section, the authors can consider mentioning the characteristics of the three included studies.

Demographic characteristics have been added to abstract

Abstract

Keywords:

2.         remove FuCT

Keywords FuCT has been removed

keywords

Introduction:

3.         The length of the introduction is appropriate, but the description of the Functional Chewing Training occupied only around 10% of the introduction. The authors can consider extending the description of this approach. For example, how long has it been used in clinical practice, whether it has been applied in other populations, and what are the examples of postural alignment and sensory and motor training?

Information about the FuCT have been added

98-101

4.         Line 38, provides a reference for the statement ‘…. malnutrition seen in 90% of this patient population

Reference has been added

References

5.         Two different ways to cite in-text references were used.

References have been corrected

Results

6.         Line 166 -167: ‘Therefore, the GMFCS may ………the homogeneity of groups’. Interpretation and suggestions should be moved to the discussion.

Sentence has been moved

7.         Line 173: I am not sure how the authors get the number mentioned in this sentence (FuCT group= 12 females/14 males; control group= 9 females/14 males)

This sentence has been removed

8.         Any study reported an adverse effect?

No studies reported a adverse effect

174

9.         Two excluded studies which have been full-text reviewed have been excluded. The authors can provide the reference of the two papers for information.

These two studies were not on CP, it has been reported in the flowchart

Figure 1

10.     Discussion

The discussion section is filled with a description of the results of the included studies (Line 222 -241). From my point of view, this information should be presented in the results section. Besides, although no meta-analyses are conducted in this review, the authors could try to calculate the effect size of each individual study to quantify the effect of the FuCT.

Some sentence has been removed

11.     The authors should try to make a more specific recommendation on the future research direction and study design for the future study. For example, the author suggested, ' It is necessary to conduct studies with a large number of participants (Line 253)’. The authors could suggest a sample size based on the studies reviewed in this manuscript.

Recommendations on the future research has been added

269-274

12.     The justification for using GMFCS in future studies seems very weak to me. The authors should try to justify their recommendation based on the tool's strength and validity and whether the tool can reflect the intended treatment effect instead of based on the fact that 2 of the included studies using it.

This information has been added

282,283

13.     The authors suggest using the ‘Eating and Drinking Ability Classification System’ in future study, but the rationale has never been mentioned in the discussion session.

The rationale has been mentioned in the discussion section

230-231

Conclusion:

14.     Some suggestions in the conclusion section have not been or have just briefly discussed. The authors should provide stronger justification for the suggestions.

Stroger justifications for the suggestions have been added to discussion

Discussion secion

15.     Line 271 ‘it is recommended to evaluate a priori the condition of subjects with epilepsy associated with CP’, However, only one study reported dropout due to epilepsy-associated conditions according to the results presented in this study. Perhaps the authors can try to explain why children with epilepsy-associated conditions tend to drop out from the training and why this did not happen in the other two studies.

279-280

16. Figure 1: Some words are missing from the figure

Figure has been revised

Figure 1

We hope that the new version of our manuscript is acceptable for publication.

Best regards,

Giovanni Galeoto

Reviewer 2 Report

The manuscript provides a useful summary of recent studies on whether Functional Chewing Training is beneficial to children with cerebral palsy. The meta-analysis shows FuCT consistently improves chewing functions and related defects across 3 studies that are selected based on a set of quality criteria. The conclusions are reassuring and provide high-level support for implementing FuCT in clinical practices.

In the meantime, the manuscript's analysis and conclusions are too straightforward to qualify as a new publication. After selecting a few studies to analyze, the manuscript essentially re-iterates the results from previous studies and lacks deeper insights. I would recommend the authors conduct a deeper analysis of the questions below: 

1. The authors summarized the statistically significant chewing performance improvements in the 3 studies, but did not discuss the magnitude of improvements (is it a very small improvement, or a big enough improvement to make the children's life much better?). The authors should also state the KCPS scores (and other scores) in trial and control groups, and what the change means for the patient's life quality. 

2. Does the magnitude change differ across the 3 studies?

Additional comments:

1. The authors stated "It is necessary to conduct studies with a large number of participants so that conclusive results and sufficient scientific evidence are obtained." Why do we need larger-size studies when 3 independent studies already give consistent and statistically significant conclusions? I would argue that even with some limitations of the previous work, 3 medium-to-high-quality studies (as evaluated by Jadad and PEDro) showing the same result means we don't need to spend more funding resources to repeat the same kind of study. Regarding the limitations of the 3 studies, small sample size is not an issue if the study can draw statistically significant conclusions (i.e. it already has sufficient statistical power). 

2. The statement "There is insufficient scientific evidence to compare FuCT effectiveness in children with CP." is not consistent with the result of the meta-analysis.

3. The authors should include demographic information of the 3 studies and their participants.

Author Response

(The authors gave the same response as above.)

Reviewer 3 Report

Thanks for the invitation to review the manuscript ' Evaluation of the effectiveness of Functional Chewing Training compared to standard treatment in a population of children with cerebral palsy: A systematic review of randomized controlled trials'.

My major concern is that the results of this review have not been thoroughly discussed. And recommendations for future study are too brief and lack strong justification.

Below please find my comments and suggestions.

Abstract:

1.         Instead of mentioning how many studies were screened and removed in each step in the results section, the authors can consider mentioning the characteristics of the three included studies.

Keywords:

2.         remove FuCT

Introduction:

3.         The length of the introduction is appropriate, but the description of the Functional Chewing Training occupied only around 10% of the introduction. The authors can consider extending the description of this approach. For example, how long has it been used in clinical practice, whether it has been applied in other populations, and what are the examples of postural alignment and sensory and motor training?

4.         Line 38, provides a reference for the statement ‘…. malnutrition seen in 90% of this patient population

5.         Two different ways to cite in-text references were used.

Results

6.         Line 166 -167: ‘Therefore, the GMFCS may ………the homogeneity of groups’. Interpretation and suggestions should be moved to the discussion.

7.         Line 173: I am not sure how the authors get the number mentioned in this sentence (FuCT group= 12 females/14 males; control group= 9 females/14 males)

8.         Any study reported an adverse effect?

9.         Two excluded studies which have been full-text reviewed have been excluded. The authors can provide the reference of the two papers for information.

10.     Discussion

The discussion section is filled with a description of the results of the included studies (Line 222 -241). From my point of view, this information should be presented in the results section. Besides, although no meta-analyses are conducted in this review, the authors could try to calculate the effect size of each individual study to quantify the effect of the FuCT.

11.     The authors should try to make a more specific recommendation on the future research direction and study design for the future study. For example, the author suggested, ' It is necessary to conduct studies with a large number of participants (Line 253)’. The authors could suggest a sample size based on the studies reviewed in this manuscript.

12.     The justification for using GMFCS in future studies seems very weak to me. The authors should try to justify their recommendation based on the tool's strength and validity and whether the tool can reflect the intended treatment effect instead of based on the fact that 2 of the included studies using it.

13.     The authors suggest using the ‘Eating and Drinking Ability Classification System’ in future study, but the rationale has never been mentioned in the discussion session.

Conclusion:

14.     Some suggestions in the conclusion section have not been or have just briefly discussed. The authors should provide stronger justification for the suggestions.

15.     Line 271 ‘it is recommended to evaluate a priori the condition of subjects with epilepsy associated with CP’, However, only one study reported dropout due to epilepsy-associated conditions according to the results presented in this study. Perhaps the authors can try to explain why children with epilepsy-associated conditions tend to drop out from the training and why this did not happen in the other two studies.

16. Figure 1: Some words are missing from the figure

Author Response

(The authors gave the same response as above.)

Round 2

Reviewer 3 Report

The authors have addressed my comments and I am satisfied with the quality of the revised manuscript.